# Principal Stress Trajectories in Plasticity under Plane Strain and Axial Symmetry

Sergei Alexandrov [1,2,3], Marina Rynkovskaya [3] and Yong Li [2,*]

1 Ishlinsky Institute for Problems in Mechanics RAS, 101-1 Prospect Vernadskogo, 119526 Moscow, Russia
2 School of Mechanical Engineering and Automation, Beihang University, No. 37 Xueyuan Road, Beijing 100191, China
3 Department of Civil Engineering, Peoples' Friendship University of Russia (RUDN University), 6 Miklukho-Maklaya St., 117198 Moscow, Russia
* Correspondence: liyong19@buaa.edu.cn

**Abstract:** The two families of principal stress trajectories can be regarded as an orthogonal curvilinear coordinate system under plane strain and axial symmetry. Under plane strain, the equilibrium equations in conjunction with a yield criterion comprise a statically determinate system. Under axial symmetry, a statically determinate system results from the above equations supplemented with the hypothesis of Haar von Karman. In both cases, the compatibility equations for mapping the principal line coordinate system to a given coordinate system show that the scale factors of the former satisfy a simple algebraic or transcendental equation for many yield criteria. Using this equation, one can develop a method for reducing boundary value problems in plasticity to purely geometric problems. The method is independent of any flow rule that can be chosen to calculate displacement or velocity fields, as well as independent whether elastic strains are included. The present paper summarizes available results related to using principal stress trajectories in plasticity and emphasizes the advantages of the method above.

**Keywords:** principal stress trajectories; scale factors; yield criteria; plasticity

## 1. Introduction

In the case of plane strain and axisymmetric deformations, principal stress trajectories can be regarded as the coordinate curves of an orthogonal coordinate system. This coordinate system is named the principal line coordinate system. Typical constitutive equations of plasticity include a yield criterion. In the case of isotropic perfectly plastic solids, the yield criterion is an equation that involves stress invariants and constitutive parameters. Depending on the yield criterion, the principal line coordinate system has remarkable geometric properties in regions where the yield criterion is satisfied. The pioneering work that discovered such properties was published in 1941 [1]. This paper has considered the plane strain deformation of solids obeying Tresca's yield criterion. The derivation is also valid for an arbitrary pressure-independent yield criterion if the material is regarded as rigid plastic. The present paper reviews the extension of the result reported in [1] to more general yield criteria and axisymmetric deformation. The phrase 'linear yield criterion' means that the yield criterion is represented by a linear function of the principal stresses. The theory has been completed for such yield criteria. The research on more general yield criteria is ongoing. The first results are included in the present paper.

Besides geometric properties, various aspects of research related to principal stress trajectories have been reported in the literature. Paper [2] has developed a principal line theory of axisymmetric plastic deformation, assuming that a face regime of Tresca's yield criterion is operative. It has been emphasized in [3] that using principal line coordinates proves advantageous for solving boundary value problems, including numerically. The practical usefulness of principal line coordinates has also been noted in [4].

The Ideal flow theory is a tool for designing metal-forming processes [5]. A property of stationary bulk ideal flows is that principal stress trajectories coincide with streamlines [6,7]. A property of non-stationary bulk ideal flows is that principal stress trajectories are material lines [8]. In both cases, calculating ideal flows is equivalent to calculating principal stress trajectories.

## 2. Basic Equations

### 2.1. Coordinate Systems

The principal stress directions are orthogonal. Therefore, two families of the principal stress trajectories can be regarded as a curvilinear orthogonal system in a generic flow plane under plane strain. This coordinate system is denoted as $(\xi,\ \eta)$. Besides, a Cartesian coordinate system in the same plane will be used. This coordinate system is denoted as $(x,\ y)$. These two coordinate systems are depicted in Figure 1, where $\psi$ is the orientation of the $\xi$-lines relative to the $x$-axis, measured anticlockwise positive from the $x$-axis. The scale factors of the $\xi$- and $\eta$-coordinate lines are denoted as $h_\xi$ and $h_\eta$, respectively. It is seen from Figure 1 that

$$\frac{\partial x}{\partial \xi} = h_\xi \cos\psi, \quad \frac{\partial x}{\partial \eta} = -h_\eta \sin\psi, \quad \frac{\partial y}{\partial \xi} = h_\xi \sin\psi, \quad \frac{\partial y}{\partial \eta} = h_\eta \cos\psi. \tag{1}$$

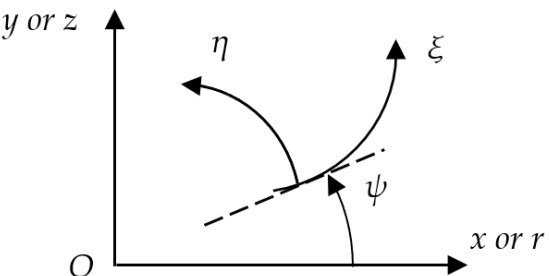

**Figure 1.** Principal line coordinate system.

The compatibility equations are

$$\frac{\partial^2 x}{\partial \xi \partial \eta} = \frac{\partial^2 x}{\partial \eta \partial \xi} \quad \text{and} \quad \frac{\partial^2 y}{\partial \xi \partial \eta} = \frac{\partial^2 y}{\partial \eta \partial \xi}. \tag{2}$$

Equations (1) and (2) combine to give

$$\begin{aligned} \frac{\partial h_\xi}{\partial \eta}\cos\psi - h_\xi \sin\psi \frac{\partial \psi}{\partial \eta} + \frac{\partial h_\eta}{\partial \xi}\sin\psi + h_\eta \cos\psi \frac{\partial \psi}{\partial \xi} &= 0, \\ \frac{\partial h_\xi}{\partial \eta}\sin\psi + h_\xi \cos\psi \frac{\partial \psi}{\partial \eta} - \frac{\partial h_\eta}{\partial \xi}\cos\psi + h_\eta \sin\psi \frac{\partial \psi}{\partial \xi} &= 0. \end{aligned} \tag{3}$$

Multiplying the first equation by $\cos\psi$, the second by $\sin\psi$ and summing gives

$$\frac{\partial h_\xi}{\partial \eta} + h_\eta \frac{\partial \psi}{\partial \xi} = 0. \tag{4}$$

Similarly, multiplying $(3)^1$ by $-\sin\psi$, $(3)^2$ by $\cos\psi$ and summing gives

$$h_\xi \frac{\partial \psi}{\partial \eta} - \frac{\partial h_\eta}{\partial \xi} = 0. \tag{5}$$

Under axial symmetry, two families of the principal stress trajectories can be regarded as a curvilinear orthogonal system in a generic meridian plane. This coordinate system is denoted as $(\xi,\ \theta,\ \eta)$. Besides, a cylindrical coordinate system $(r,\ \theta,\ z)$ will be used. In a generic meridian plane, these two coordinate systems are depicted in Figure 1, where $\psi$ is the orientation of the $\xi$-lines relative to the $r$-axis, measured anticlockwise positive from

the *r*-axis. The $(\xi, \theta, \eta)$ coordinate system is a principal line coordinate system. The scale factors of the $\xi$- and $\eta$-coordinate curves are denoted as $h_\xi$ and $h_\eta$, respectively. The scale factor of the $\theta$-coordinate lines is *r*. It is seen from Figure 1 that

$$\frac{\partial r}{\partial \xi} = h_\xi \cos \psi, \quad \frac{\partial r}{\partial \eta} = -h_\eta \sin \psi, \quad \frac{\partial z}{\partial \xi} = h_\xi \sin \psi, \quad \frac{\partial z}{\partial \eta} = h_\eta \cos \psi. \tag{6}$$

The same line of reasoning that has led to Equations (4) and (5), starting from Equation (1), can be used to show that (4) and (5) are valid under axial symmetry.

### 2.2. Plasticity

The models considered herein are perfectly plastic and statically determinate. The latter means that the plastic flow rule and elastic deformation are immaterial for analyzing the stress equations. The system of equations consists of the equilibrium equations and a yield (or failure, or strength) criterion. The term yield will be used herein.

Let $\sigma_\xi$ and $\sigma_\eta$ be the normal stresses referred to the principal line coordinate system. These stresses are principal stresses. Under axial symmetry, the third principal stress is denoted as $\sigma_\theta$. The equilibrium equations are [9]

$$h_\eta \frac{\partial \sigma_\xi}{\partial \xi} + (\sigma_\xi - \sigma_\eta) \frac{\partial h_\eta}{\partial \xi} = 0 \quad \text{and} \quad h_\xi \frac{\partial \sigma_\eta}{\partial \eta} + (\sigma_\eta - \sigma_\xi) \frac{\partial h_\xi}{\partial \eta} = 0 \tag{7}$$

under plane strain and

$$\begin{gathered} \frac{\partial \sigma_\xi}{\partial \xi} + (\sigma_\xi - \sigma_\eta) \frac{\partial h_\eta}{h_\eta \partial \xi} + (\sigma_\xi - \sigma_\theta) \frac{\partial r}{r \partial \xi} = 0 \text{ and} \\ \frac{\partial \sigma_\eta}{\partial \eta} + (\sigma_\eta - \sigma_\xi) \frac{\partial h_\xi}{h_\xi \partial \eta} + (\sigma_\eta - \sigma_\theta) \frac{\partial r}{r \partial \eta} = 0 \end{gathered} \tag{8}$$

under axial symmetry.

Any isotropic plane strain yield criterion can be represented as

$$\Phi_p (\sigma_\xi, \sigma_\eta) = \sigma_0. \tag{9}$$

Here, $\Phi_p (\sigma_\xi, \sigma_\eta)$ is an arbitrary function of its arguments satisfying the standard requirements of plasticity theory and is $\sigma_0$ a reference stress. Equations (4), (5), (7), and (9) form a determinate system for $h_\xi$, $h_\eta$, $\psi$, $\sigma_\xi$, and $\sigma_\eta$. Under axial symmetry, piece-wise differentiable yield criteria are considered. A typical yield criterion can be represented as

$$\Phi_a^{(1)} (\sigma_\xi, \sigma_\theta, \sigma_\eta) = \sigma_0 \quad \text{and} \quad \Phi_a^{(2)} (\sigma_\xi, \sigma_\theta, \sigma_\eta) = \sigma_0. \tag{10}$$

It is assumed that the equations in (10) can be rewritten as

$$\Phi_a (\sigma_\xi, \sigma_\eta) = \sigma_0 \tag{11}$$

and

$$\sigma_\xi = \sigma_\theta \text{ or } \sigma_\eta = \sigma_\theta. \tag{12}$$

The last equation expresses the hypothesis of Haar von Karman. Equations (4), (5), (8), (11), and (12) form a determinate system for $h_\xi$, $h_\eta$, $\psi$, $\sigma_\xi$, $\sigma_\eta$, and $\sigma_\theta$.

In what follows, it is assumed without loss of generality that

$$\sigma_\xi > \sigma_\eta. \tag{13}$$

### 3. Relations between $h_\xi$ and $h_\eta$ under Plane Strain

*3.1. Tresca Yield Criterion*

Tresca's yield criterion is often used in metal plasticity [10]. Considering (13), this yield criterion can be represented as

$$\sigma_\xi - \sigma_\eta = 2k, \tag{14}$$

where $k$ is the shear yield stress. Eliminating $\sigma_\xi - \sigma_\eta$ in (7) using (14), one gets

$$h_\eta \frac{\partial \sigma_\xi}{\partial \xi} + 2k \frac{\partial h_\eta}{\partial \xi} = 0 \quad \text{and} \quad h_\xi \frac{\partial \sigma_\eta}{\partial \eta} - 2k \frac{\partial h_\xi}{\partial \eta} = 0. \tag{15}$$

Each of these equations can be immediately integrated to give

$$\sigma_\xi = -2k \ln h_\eta + 2kC_1(\eta) \quad \text{and} \quad \sigma_\eta = 2k \ln h_\xi + 2kC_2(\xi). \tag{16}$$

Here, $C_1(\eta)$ and $C_2(\xi)$ are arbitrary functions of $\eta$ and $\xi$, respectively. However, different choices of these functions merely change the scale of the coordinate curves. Therefore, without loss of generality, it is possible to choose $C_1(\eta) = 0$ and $C_2(\xi) = -1$. Then, Equation (16) becomes

$$\sigma_\xi = -2k \ln h_\eta \quad \text{and} \quad \sigma_\eta = 2k \ln h_\xi - 2k. \tag{17}$$

Substituting (17) into (14) yields

$$h_\xi h_\eta = 1. \tag{18}$$

This relation between the scale factors has been derived in [1].

*3.2. Mohr-Coulomb Yield Criterion*

The Mohr–Coulomb yield criterion is widely used for describing the deformation of granular materials [11–13]. Considering (13), this yield criterion can be represented as

$$-p \sin \phi + q = k \cos \phi, \tag{19}$$

where

$$p = -\frac{\sigma_\xi + \sigma_\eta}{2}, \quad q = \frac{\sigma_\xi - \sigma_\eta}{2}, \tag{20}$$

$k$ is the cohesion, and $\phi$ is the angle of internal friction. Using (20), one can transform the equations in (7) to

$$h_\eta \left( \frac{\partial q}{\partial \xi} - \frac{\partial p}{\partial \xi} \right) + 2q \frac{\partial h_\eta}{\partial \xi} = 0 \quad \text{and} \quad h_\xi \left( \frac{\partial p}{\partial \eta} + \frac{\partial q}{\partial \eta} \right) + 2q \frac{\partial h_\xi}{\partial \eta} = 0. \tag{21}$$

Eliminating $p$ in these equations employing (19) yields

$$h_\eta \left( 1 - \frac{1}{\sin \phi} \right) \frac{\partial q}{\partial \xi} + 2q \frac{\partial h_\eta}{\partial \xi} = 0 \quad \text{and} \quad h_\xi \frac{\partial q}{\partial \eta} \left( 1 + \frac{1}{\sin \phi} \right) + 2q \frac{\partial h_\xi}{\partial \eta} = 0. \tag{22}$$

Each of these equations can be immediately integrated to give

$$\frac{1}{2} \left( 1 - \frac{1}{\sin \phi} \right) \ln q = -\ln h_\eta + C_1(\eta) \quad \text{and} \quad \frac{1}{2} \left( 1 + \frac{1}{\sin \phi} \right) \ln q = -\ln h_\xi + C_2(\xi). \tag{23}$$

Here, $C_1(\eta)$ and $C_2(\xi)$ are arbitrary functions of $\eta$ and $\xi$, respectively. However, different choices of these functions merely change the scale of the coordinate curves.

Therefore, without loss of generality, it is possible to choose $C_1(\eta) = 0$ and $C_2(\xi) = 0$. Then, Equation (23) becomes

$$\frac{1}{2}\left(1 - \frac{1}{\sin\phi}\right)\ln q = -\ln h_\eta \quad \text{and} \quad \frac{1}{2}\left(1 + \frac{1}{\sin\phi}\right)\ln q = -\ln h_\xi. \tag{24}$$

Eliminating $q$ between these equations results in

$$h_\xi^m h_\eta = 1, \tag{25}$$

where $m = (1 - \sin\phi)/(1 + \sin\phi)$. This equation reduces to (18) at $\phi = 0$.

### 3.3. Generalized Linear Yield Criterion

Generalized linear yield criteria are often adopted in the literature [14–19]. Considering (13), a typical generalized linear yield criterion can be represented as

$$\sigma_\eta = t\sigma_\xi - \sigma_0, \tag{26}$$

where $t > 0$ is a constitutive parameter. This equation reduces to (14) at $t = 1$. Therefore, it is assumed in this section that $t \neq 1$. Eliminating $\sigma_\eta$ in (7) using (26), one gets

$$h_\eta \frac{\partial \sigma_\xi}{\partial \xi} + [\sigma_\xi(1-t) + \sigma_0]\frac{\partial h_\eta}{\partial \xi} = 0 \text{ and } -[\sigma_\xi(1-t) + \sigma_0]\frac{\partial h_\xi}{\partial \eta} + th_\xi\frac{\partial \sigma_\xi}{\partial \eta} = 0. \tag{27}$$

Integrating these equations leads to

$$\begin{aligned} \frac{1}{(1-t)}\ln\left[\frac{\sigma_\xi}{\sigma_0}(1-t)+1\right] &= -\ln h_\eta + C_1(\eta) \text{ and} \\ \frac{t}{(1-t)}\ln\left[\frac{\sigma_\xi}{\sigma_0}(1-t)+1\right] &= \ln h_\xi + C_2(\xi). \end{aligned} \tag{28}$$

Here, $C_1(\eta)$ and $C_2(\xi)$ are arbitrary functions of $\eta$ and $\xi$, respectively. However, different choices of these functions merely change the scale of the coordinate curves. Therefore, without loss of generality, it is possible to choose $C_1(\eta) = C_2(\xi) = 0$. Then, Equation (28) becomes

$$\frac{1}{(1-t)}\ln\left[\frac{\sigma_\xi}{\sigma_0}(1-t)+1\right] = -\ln h_\eta \text{ and } \frac{t}{(1-t)}\ln\left[\frac{\sigma_\xi}{\sigma_0}(1-t)+1\right] = \ln h_\xi. \tag{29}$$

Eliminating $\sigma_\xi$ between these equations results in

$$h_\xi^{1/t} h_\eta = 1. \tag{30}$$

This relation between the scale factors was derived in [19]. Equation (30) reduces to (18) at $t = 1$.

### 3.4. General Yield Criterion

An adequate description of some materials requires non-linear yield criteria [19–22]. It is convenient to represent (9) as

$$q = f(p), \tag{31}$$

where $q$ and $p$ are defined as in (20). Expressing the principal stresses in terms of $p$ and $q$, eliminating $q$ by employing (31), and substituting the resulting expressions in the equations in (7) leads to

$$\begin{aligned} \frac{\partial h_\eta}{h_\eta \partial \xi} + \frac{1}{2}\left[\frac{1}{f(p)}\frac{df}{dp} - \frac{1}{f(p)}\right]\frac{\partial p}{\partial \xi} &= 0, \\ \frac{\partial h_\xi}{h_\xi \partial \eta} + \frac{1}{2}\left[\frac{1}{f(p)}\frac{df}{dp} + \frac{1}{f(p)}\right]\frac{\partial p}{\partial \eta} &= 0. \end{aligned} \tag{32}$$

Each of these equations can be immediately integrated to give

$$\ln h_\eta + C_1(\eta) = \tfrac{1}{2} \int_0^p \frac{d\chi}{f(\chi)} - \tfrac{1}{2} \ln\left[\frac{f(p)}{f(0)}\right],$$
$$\ln h_\xi + C_2(\xi) = -\tfrac{1}{2} \int_0^p \frac{d\chi}{f(\chi)} - \tfrac{1}{2} \ln\left[\frac{f(p)}{f(0)}\right]. \tag{33}$$

Here, $C_1(\eta)$ and $C_2(\xi)$ are arbitrary functions of $\eta$ and $\xi_{,,}$ respectively. However, different choices of these functions merely change the scale of the coordinate curves. Therefore, without loss of generality, it is possible to choose $C_1(\eta) = C_2(\xi) = 0$. Then, Equation (33) becomes

$$\ln h_\eta = \tfrac{1}{2} \int_0^p \frac{d\chi}{f(\chi)} - \tfrac{1}{2} \ln\left[\frac{f(p)}{f(0)}\right],$$
$$\ln h_\xi = -\tfrac{1}{2} \int_0^p \frac{d\chi}{f(\chi)} - \tfrac{1}{2} \ln\left[\frac{f(p)}{f(0)}\right]. \tag{34}$$

These equations supply the relation between $h_\xi$ and $h_\eta$ in parametric form with $p$ being the parameter. This relation was derived in [23]. It has been shown in this work that (34) reduces to (18), (25), or (30) if the function $f$ in (31) is chosen accordingly.

## 4. Relations between $h_\xi$ and $h_\eta$ under Axial Symmetry

### 4.1. Tresca Yield Criterion

Considering (13), Equation (11) in the case of Tresca's yield criterion becomes

$$\sigma_\xi - \sigma_\eta = 2k. \tag{35}$$

Eliminating $\sigma_\xi - \sigma_\eta$ and $\sigma_\theta$ in (8) using (12)[1] and (14), one gets

$$h_\eta \frac{\partial \sigma_\xi}{\partial \xi} + 2k \frac{\partial h_\eta}{\partial \xi} = 0 \quad \text{and} \quad \frac{\partial \sigma_\eta}{\partial \eta} - 2k\left(\frac{\partial h_\xi}{h_\xi \partial \eta} + \frac{\partial r}{r \partial \eta}\right) = 0. \tag{36}$$

Each of these equations can be immediately integrated to give

$$\sigma_\xi = -2k \ln h_\eta + 2kC_1(\eta) \quad \text{and} \quad \sigma_\eta = 2k \ln(rh_\xi) + 2kC_2(\xi). \tag{37}$$

Here, $C_1(\eta)$ and $C_2(\xi)$ are arbitrary functions of $\eta$ and $\xi$, respectively. However, different choices of these functions merely change the scale of the coordinate curves. Therefore, without loss of generality it is possible to choose $C_1(\eta) = 0$ and $C_2(\xi) = -1$. Then, Equation (37) becomes

$$\sigma_\xi = -2k \ln h_\eta \quad \text{and} \quad \sigma_\eta = 2k \ln(rh_\xi) - 2k. \tag{38}$$

Substituting (38) into (35) yields

$$rh_\xi h_\eta = 1. \tag{39}$$

This relation between the scale factors was derived in [24]. Using the line of reasoning above, one can derive Equation (39) employing (12)[2] instead of (12)[1].

### 4.2. Mohr-Coulomb Yield Criterion

Considering (13), Equation (11) in the case of the Coulomb–Mohr yield criterion becomes

$$-p \sin \phi + q = k \cos \phi, \tag{40}$$

Using (12)[1] and (20), one can transform Equation (8) to

$$h_\eta \left( \frac{\partial q}{\partial \xi} - \frac{\partial p}{\partial \xi} \right) + 2q \frac{\partial h_\eta}{\partial \xi} = 0 \quad \text{and} \quad \frac{\partial p}{\partial \eta} + \frac{\partial q}{\partial \eta} + 2q \left( \frac{\partial h_\xi}{h_\xi \partial \eta} + \frac{\partial h_\xi}{r \partial \eta} \right) = 0. \tag{41}$$

Eliminating $p$ in these equations employing (40) yields

$$h_\eta \left( 1 - \frac{1}{\sin\phi} \right) \frac{\partial q}{\partial \xi} + 2q \frac{\partial h_\eta}{\partial \xi} = 0 \quad \text{and} \quad \frac{\partial q}{\partial \eta} \left( 1 + \frac{1}{\sin\phi} \right) + 2q \left( \frac{\partial h_\xi}{h_\xi \partial \eta} + \frac{\partial h_\xi}{r \partial \eta} \right) = 0. \tag{42}$$

Each of these equations can be immediately integrated to give

$$\frac{1}{2} \left( 1 - \frac{1}{\sin\phi} \right) \ln q = -\ln h_\eta + C_1(\eta) \quad \text{and} \quad \frac{1}{2} \left( 1 + \frac{1}{\sin\phi} \right) \ln q = -\ln(rh_\xi) + C_2(\xi). \tag{43}$$

Here, $C_1(\eta)$ and $C_2(\xi)$ are arbitrary functions of $\eta$ and $\xi$, respectively. However, different choices of these functions merely change the scale of the coordinate curves. Therefore, without loss of generality, it is possible to choose $C_1(\eta) = 0$ and $C_2(\xi) = 0$. Then, Equation (43) becomes

$$\frac{1}{2} \left( 1 - \frac{1}{\sin\phi} \right) \ln q = -\ln h_\eta \quad \text{and} \quad \frac{1}{2} \left( 1 + \frac{1}{\sin\phi} \right) \ln q = -\ln(rh_\xi). \tag{44}$$

Eliminating $q$ between these equations results in

$$(rh_\xi)^m h_\eta = 1. \tag{45}$$

The definition of $m$ is provided after Equation (25).

Employing (12)[2] instead of (12)[1] and using the line of reasoning above, one gets

$$rh_\eta h_\xi^m = 1. \tag{46}$$

### 4.3. Generalized Linear Yield Criterion

Considering (13), Equation (11) in the case of a typical generalized linear yield criterion becomes

$$\sigma_\eta = t\sigma_\xi - \sigma_0, \tag{47}$$

As in Section 3.3, it is assumed in this section that $t \neq 1$. Eliminating $\sigma_\eta$ and $\sigma_\theta$ in (8) using (12)[1] and (47), one gets

$$\frac{1}{[\sigma_\xi(1-t) + \sigma_0]} \frac{\partial \sigma_\xi}{\partial \xi} + \frac{\partial h_\eta}{h_\eta \partial \xi} = 0 \text{ and } -\frac{t}{[\sigma_\xi(1-t) + \sigma_0]} \frac{\partial \sigma_\eta}{\partial \eta} + \frac{1}{rh_\xi} \frac{\partial(rh_\xi)}{\partial \eta} = 0. \tag{48}$$

Integrating these equations leads to

$$\frac{\sigma_\xi}{\sigma_0} = \frac{\left[ 1 - C_1(\eta) h_\eta^{t-1} \right]}{t-1} \text{ and } \frac{\sigma_\eta}{\sigma_0} = \frac{\left[ 1 - C_2(\xi)(rh_\xi)^{(1-t)/t} \right]}{t-1}. \tag{49}$$

Here, $C_1(\eta)$ and $C_2(\xi)$ are arbitrary functions of $\eta$ and $\xi$, respectively. However, different choices of these functions merely change the scale of the coordinate curves. Therefore, without loss of generality, it is possible to choose $C_1(\eta) = C_2(\xi)/t = 1$. Then, Equation (49) becomes

$$\frac{\sigma_\xi}{\sigma_0} = \frac{\left( 1 - h_\eta^{t-1} \right)}{t-1} \text{ and } \frac{\sigma_\eta}{\sigma_0} = \frac{\left[ 1 - t(rh_\xi)^{(1-t)/t} \right]}{t-1}. \tag{50}$$

Substituting (50) into (47) results in

$$\left(rh_\xi\right)^{1/t} h_\eta = 1. \tag{51}$$

Employing $(12)^2$ instead of $(12)^1$ and using the line of reasoning above, one gets

$$rh_\eta h_\xi^{1/t} = 1. \tag{52}$$

Equations (51) and (52) have been derived in [25].

## 5. Final Equation Systems under Plane Strain

### 5.1. Linear Yield Criteria

Equations (18), (25), and (30) can be represented as

$$h_\xi^{-\tau} h_\eta = 1, \tag{53}$$

where $\tau = -1$ in Equation (18), $\tau = -m$ in Equation (25), and $\tau = -1/t$ in Equation (30). Equation (53) can be rewritten in the form

$$h_\xi = h \quad \text{and} \quad h_\eta = h^\tau. \tag{54}$$

Then, Equations (4) and (5) become

$$\frac{\partial h}{\partial \eta} + h^\tau \frac{\partial \psi}{\partial \xi} = 0 \quad \text{and} \quad \frac{\partial \psi}{\partial \eta} - \tau h^{\tau-2} \frac{\partial h}{\partial \xi} = 0. \tag{55}$$

Using a standard technique, one can find the characteristic curves as

$$\frac{d\xi}{d\eta} = \pm\sqrt{-\tau} h^{\tau-1}. \tag{56}$$

It is seen from this equation that the characteristics are real if $\tau < 0$. The hyperbolic regimes are most important in perfectly plastic solids [26]. Therefore, it is assumed that $\tau < 0$ in the reminder of the present paper. Note that this inequality is automatically satisfied in the case of Equations (18) and (25). In the case of Equation (30), the inequality $\tau < 0$ is equivalent to $t > 0$.

The characteristic coordinates are denoted as $(\alpha, \beta)$. The upper sign in (56) corresponds to the $\beta$-curves and the lower sign to the $\alpha$-curves. Using a standard technique, one can find the characteristic relations as

$$\begin{aligned} d\psi - \sqrt{-\tau}\frac{dh}{h} = 0 \text{ along the } \alpha - \text{lines,} \\ d\psi + \sqrt{-\tau}\frac{dh}{h} = 0 \text{ along the } \beta - \text{lines.} \end{aligned} \tag{57}$$

These equations can immediately be integrated to give

$$\psi - \sqrt{-\tau}\ln h = 2g_1(\beta) \quad \text{and} \quad \psi + \sqrt{-\tau}\ln h = 2g_2(\alpha). \tag{58}$$

Here, $g_1(\beta)$ is an arbitrary function only of $\beta$ and $g_2(\alpha)$ is an arbitrary function only of $\alpha$. The equations in (58) can be solved for $\psi$ and $h$. As a result,

$$\psi = g_1(\beta) + g_2(\alpha) \quad \text{and} \quad \sqrt{-\tau}\ln h = g_2(\alpha) - g_1(\beta). \tag{59}$$

If both families of the characteristic lines are curved, one can choose $g_1(\beta) = c\sqrt{-\tau}\beta$ and $g_2(\alpha) = c\sqrt{-\tau}\alpha$, where $c$ is constant. Then, the equations in (59) become

$$\psi = c\sqrt{-\tau}(\alpha + \beta) \quad \text{and} \quad \ln h = c(\alpha - \beta). \tag{60}$$

The equations in (56) can be rewritten as

$$\frac{\partial \zeta}{\partial \alpha} + \sqrt{-\tau}h^{\tau-1}\frac{\partial \eta}{\partial \alpha} = 0 \quad \text{and} \quad \frac{\partial \zeta}{\partial \beta} - \sqrt{-\tau}h^{\tau-1}\frac{\partial \eta}{\partial \beta} = 0. \tag{61}$$

Eliminating $h$ in these equations using the second equation in (60) yields

$$\begin{array}{c}\frac{\partial \zeta}{\partial \alpha} + \sqrt{-\tau}\exp[c(\tau-1)(\alpha-\beta)]\frac{\partial \eta}{\partial \alpha} = 0 \text{ and} \\ \frac{\partial \zeta}{\partial \beta} - \sqrt{-\tau}\exp[c(\tau-1)(\alpha-\beta)]\frac{\partial \eta}{\partial \beta} = 0.\end{array} \tag{62}$$

It is convenient to introduce the new quantities, $\varepsilon$ and $\nu$, as

$$\varepsilon = \zeta\exp[n(\beta-\alpha)] \quad \text{and} \quad \nu = \eta\exp[n(\alpha-\beta)], \tag{63}$$

where $n$ is constant. Differentiating these expressions with respect to $\alpha$ and $\beta$, one gets

$$\begin{array}{cc}\frac{\partial \zeta}{\partial \alpha} = \left(\frac{\partial \varepsilon}{\partial \alpha} + \varepsilon n\right)\exp[n(\alpha-\beta)], & \frac{\partial \zeta}{\partial \beta} = \left(\frac{\partial \varepsilon}{\partial \beta} - \varepsilon n\right)\exp[n(\alpha-\beta)], \\ \frac{\partial \eta}{\partial \alpha} = \left(\frac{\partial \nu}{\partial \alpha} - \nu n\right)\exp[n(\beta-\alpha)], & \frac{\partial \eta}{\partial \beta} = \left(\frac{\partial \nu}{\partial \beta} + \nu n\right)\exp[n(\beta-\alpha)].\end{array} \tag{64}$$

Equations (62) and (64) combine to give

$$\begin{array}{c}\left(\frac{\partial \varepsilon}{\partial \alpha} + \varepsilon n\right)\exp[n(\alpha-\beta)] + \sqrt{-\tau}\exp[m(\tau-1)(\alpha-\beta)]\exp[n(\beta-\alpha)]\left(\frac{\partial \nu}{\partial \alpha} - \nu n\right) = 0, \\ \left(\frac{\partial \varepsilon}{\partial \beta} - \varepsilon n\right)\exp[n(\alpha-\beta)] - \sqrt{-\tau}\exp[m(\tau-1)(\alpha-\beta)]\exp[n(\beta-\alpha)]\left(\frac{\partial \nu}{\partial \beta} + \nu n\right) = 0.\end{array} \tag{65}$$

Put

$$c = \frac{2n}{\tau-1}. \tag{66}$$

Then, the equations in (65) become

$$\left(\frac{\partial \varepsilon}{\partial \alpha} + \varepsilon n\right) + \sqrt{-\tau}\left(\frac{\partial \nu}{\partial \alpha} - \nu n\right) = 0 \text{ and } \left(\frac{\partial \varepsilon}{\partial \beta} - \varepsilon n\right) - \sqrt{-\tau}\left(\frac{\partial \nu}{\partial \beta} + \nu n\right) = 0. \tag{67}$$

It is convenient to introduce the new quantities, $\omega$ and $\mu$, as

$$\omega = \varepsilon + \sqrt{-\tau}\nu \quad \text{and} \quad \mu = \varepsilon - \sqrt{-\tau}\nu. \tag{68}$$

Then, the equations in (67) transform to

$$\frac{\partial \omega}{\partial \alpha} + n\mu = 0 \quad \text{and} \quad \frac{\partial \mu}{\partial \beta} - n\omega = 0. \tag{69}$$

Put

$$n = -1 \tag{70}$$

The equations in (69) become

$$\frac{\partial \omega}{\partial \alpha} - \mu = 0 \quad \text{and} \quad \frac{\partial \mu}{\partial \beta} + \omega = 0. \tag{71}$$

These equations are equivalent to the two telegraph equations:

$$\frac{\partial^2 \omega}{\partial \alpha \partial \beta} + \omega = 0 \quad \text{and} \quad \frac{\partial^2 \mu}{\partial \alpha \partial \beta} + \mu = 0. \tag{72}$$

Each of these equations is integrated by the method of Riemann.

Using (66) and (70), one can transform (60) and (63) to

$$\psi = \frac{2\sqrt{-\tau}}{(1-\tau)}(\alpha + \beta), \quad \ln h = \frac{2}{(1-\tau)}(\alpha - \beta),$$
$$\varepsilon = \xi \exp(\alpha - \beta), \quad \nu = \eta \exp(\beta - \alpha). \tag{73}$$

The derivatives with respect to $\alpha$ and $\beta$ can be represented as

$$\frac{\partial}{\partial \alpha} = \frac{\partial}{\partial \xi}\frac{\partial \xi}{\partial \alpha} + \frac{\partial}{\partial \eta}\frac{\partial \eta}{\partial \alpha} \quad \text{and} \quad \frac{\partial}{\partial \beta} = \frac{\partial}{\partial \xi}\frac{\partial \xi}{\partial \beta} + \frac{\partial}{\partial \eta}\frac{\partial \eta}{\partial \beta}. \tag{74}$$

Using (64) and (70), one can express the derivatives $\partial \xi/\partial \alpha$, $\partial \xi/\partial \beta$, $\partial \eta/\partial \alpha$, and $\partial \eta/\partial \beta$ as

$$\frac{\partial \xi}{\partial \alpha} = \left(\frac{\partial \varepsilon}{\partial \alpha} - \varepsilon\right)\exp(\beta - \alpha), \quad \frac{\partial \xi}{\partial \beta} = \left(\frac{\partial \varepsilon}{\partial \beta} + \varepsilon\right)\exp(\beta - \alpha),$$
$$\frac{\partial \eta}{\partial \alpha} = \left(\frac{\partial \nu}{\partial \alpha} + \nu\right)\exp(\alpha - \beta), \quad \frac{\partial \eta}{\partial \beta} = \left(\frac{\partial \nu}{\partial \beta} - \nu\right)\exp(\alpha - \beta). \tag{75}$$

Equations (1) and (54) combine to give

$$\frac{\partial x}{\partial \xi} = h\cos\psi, \quad \frac{\partial x}{\partial \eta} = -h^\tau \sin\psi, \quad \frac{\partial y}{\partial \xi} = h\sin\psi, \quad \frac{\partial y}{\partial \eta} = h^\tau \cos\psi. \tag{76}$$

Substituting (75) and (76) into (74) and eliminating $h$ employing (73) yields

$$\frac{\partial x}{\partial \alpha} = \left[\cos\psi\left(\frac{\partial \varepsilon}{\partial \alpha} - \varepsilon\right) - \sin\psi\left(\frac{\partial \nu}{\partial \alpha} + \nu\right)\right]\exp\left[\left(\frac{1+\tau}{1-\tau}\right)(\alpha - \beta)\right],$$
$$\frac{\partial x}{\partial \beta} = \left[\cos\psi\left(\frac{\partial \varepsilon}{\partial \beta} + \varepsilon\right) - \sin\psi\left(\frac{\partial \nu}{\partial \beta} - \nu\right)\right]\exp\left[\left(\frac{1+\tau}{1-\tau}\right)(\alpha - \beta)\right],$$
$$\frac{\partial y}{\partial \alpha} = \left[\sin\psi\left(\frac{\partial \varepsilon}{\partial \alpha} - \varepsilon\right) + \cos\psi\left(\frac{\partial \nu}{\partial \alpha} + \nu\right)\right]\exp\left[\left(\frac{1+\tau}{1-\tau}\right)(\alpha - \beta)\right],$$
$$\frac{\partial y}{\partial \beta} = \left[\sin\psi\left(\frac{\partial \varepsilon}{\partial \beta} + \varepsilon\right) + \cos\psi\left(\frac{\partial \nu}{\partial \beta} - \nu\right)\right]\exp\left[\left(\frac{1+\tau}{1-\tau}\right)(\alpha - \beta)\right]. \tag{77}$$

The equations in (68) can be solved for $\varepsilon$ and $\nu$. Then, utilizing (71), one can find

$$\frac{\partial \varepsilon}{\partial \alpha} - \varepsilon = \frac{1}{2}\left(\frac{\partial \mu}{\partial \alpha} - \omega\right), \quad \frac{\partial \varepsilon}{\partial \beta} + \varepsilon = \frac{1}{2}\left(\frac{\partial \omega}{\partial \beta} + \mu\right),$$
$$\frac{\partial \nu}{\partial \alpha} + \nu = -\frac{1}{2\sqrt{-\tau}}\left(\frac{\partial \mu}{\partial \alpha} - \omega\right), \quad \frac{\partial \nu}{\partial \beta} - \nu = \frac{1}{2\sqrt{-\tau}}\left(\frac{\partial \omega}{\partial \beta} + \mu\right). \tag{78}$$

Equations (77) and (78) combine to give

$$\frac{\partial x}{\partial \alpha} = \frac{1}{2}\left[\cos\psi\left(\frac{\partial \mu}{\partial \alpha} - \omega\right) + \frac{\sin\psi}{\sqrt{-\tau}}\left(\frac{\partial \mu}{\partial \alpha} - \omega\right)\right]\exp\left[\left(\frac{1+\tau}{1-\tau}\right)(\alpha - \beta)\right],$$
$$\frac{\partial x}{\partial \beta} = \frac{1}{2}\left[\cos\psi\left(\frac{\partial \omega}{\partial \beta} + \mu\right) - \frac{\sin\psi}{\sqrt{-\tau}}\left(\frac{\partial \omega}{\partial \beta} + \mu\right)\right]\exp\left[\left(\frac{1+\tau}{1-\tau}\right)(\alpha - \beta)\right],$$
$$\frac{\partial y}{\partial \alpha} = \frac{1}{2}\left[\sin\psi\left(\frac{\partial \mu}{\partial \alpha} - \omega\right) - \frac{\cos\psi}{\sqrt{-\tau}}\left(\frac{\partial \mu}{\partial \alpha} - \omega\right)\right]\exp\left[\left(\frac{1+\tau}{1-\tau}\right)(\alpha - \beta)\right],$$
$$\frac{\partial y}{\partial \beta} = \frac{1}{2}\left[\sin\psi\left(\frac{\partial \omega}{\partial \beta} + \mu\right) + \frac{\cos\psi}{\sqrt{-\tau}}\left(\frac{\partial \omega}{\partial \beta} + \mu\right)\right]\exp\left[\left(\frac{1+\tau}{1-\tau}\right)(\alpha - \beta)\right]. \tag{79}$$

In these equations, $\omega$ and $\mu$ are known functions of $\alpha$ and $\beta$ due to a solution of the equations in (72). Moreover, $\psi$ is a known function of $\alpha$ and $\beta$ due to (73). Therefore, the dependence of x and y on $\alpha$ and $\beta$ can be found by integrating (79) along any path in the $(\alpha, \beta)$-space.

*5.2. General Yield Criterion*

It is understood in this section that $h_\xi$ and $h_\eta$ are the known functions of $p$ given in the equations in (34). Then, Equations (4) and (5) become

$$\frac{dh_\xi}{dp}\frac{\partial p}{\partial \eta} + h_\eta\frac{\partial \psi}{\partial \xi} = 0 \quad \text{and} \quad h_\xi\frac{\partial \psi}{\partial \eta} - \frac{dh_\eta}{dp}\frac{\partial p}{\partial \xi} = 0. \tag{80}$$

Using a standard technique, one can find the characteristic directions as

$$\frac{d\xi}{d\eta} = \pm \exp\left[\int_0^p \frac{d\chi}{f(\chi)}\right]\sqrt{\frac{1 - df/dp}{1 + df/dp}}. \tag{81}$$

It is seen from this equation that the system is hyperbolic if

$$|df/dp| < 1. \tag{82}$$

It is assumed that this inequality is satisfied in what follows. Using a standard technique, one can find the characteristic relations as

$$d\psi + \frac{\sqrt{1 - (df/dp)^2}}{2f(p)}dp = 0 \text{ along the } \alpha - \text{lines,}$$
$$d\psi - \frac{\sqrt{1 - (df/dp)^2}}{2f(p)}dp = 0 \text{ along the } \beta - \text{lines.} \tag{83}$$

These equations can immediately be integrated to give

$$\psi + \Phi(p) = 2g_1(\beta) \quad \text{and} \quad \psi - \Phi(p) = 2g_2(\alpha), \tag{84}$$

where

$$\Phi(p) = \int_0^p \frac{\sqrt{1 - (df/d\chi)^2}}{2f(\chi)}d\chi. \tag{85}$$

The equations in (84) can be solved for $\psi$ and $\Phi$. As a result,

$$\psi = g_1(\beta) + g_2(\alpha) \quad \text{and} \quad \Phi(p) = g_1(\beta) - g_2(\alpha). \tag{86}$$

If both families of the characteristic lines are curved, one can choose $g_1(\beta) = \beta$ and $g_2(\alpha) = \alpha$. Then, the equations in (86) become

$$\psi = \alpha + \beta \quad \text{and} \quad p = \Phi^{-1}(\beta - \alpha). \tag{87}$$

Here, $\Phi^{-1}$ is the function inverse to $\Phi$. The equations in (81) can be rewritten as

$$\frac{\partial\xi}{\partial\alpha} + \exp\left[\int_0^p \frac{d\chi}{f(\chi)}\right]\sqrt{\frac{1 - df/dp}{1 + df/dp}}\frac{\partial\eta}{\partial\alpha} = 0 \text{ and}$$
$$\frac{\partial\xi}{\partial\beta} - \exp\left[\int_0^p \frac{d\chi}{f(\chi)}\right]\sqrt{\frac{1 - df/dp}{1 + df/dp}}\frac{\partial\eta}{\partial\beta} = 0. \tag{88}$$

In these equations, $p$ can be eliminated using the second equation in (87). Therefore, the equations in (88) constitute the system of two equations for $\xi$ and $\eta$.

Equations (1) and (74) combine to give

$$\frac{\partial x}{\partial\alpha} = h_\xi \cos\psi \frac{\partial\xi}{\partial\alpha} - h_\eta \sin\psi \frac{\partial\eta}{\partial\alpha}, \quad \frac{\partial x}{\partial\beta} = h_\xi \cos\psi \frac{\partial\xi}{\partial\beta} - h_\eta \sin\psi \frac{\partial\eta}{\partial\beta},$$
$$\frac{\partial y}{\partial\alpha} = h_\xi \sin\psi \frac{\partial\xi}{\partial\alpha} + h_\eta \cos\psi \frac{\partial\eta}{\partial\alpha}, \quad \frac{\partial y}{\partial\beta} = h_\xi \sin\psi \frac{\partial\xi}{\partial\beta} + h_\eta \cos\psi \frac{\partial\eta}{\partial\beta}. \tag{89}$$

One can express the right-hand sides of these equations as functions of $\alpha$ and $\beta$ employing (34) and (87). Therefore, the dependence of x and y on $\alpha$ and $\beta$ can be found by integrating (89) along any path in the ($\alpha$, $\beta$)-space.

### 6. Final Equation Systems under Axial Symmetry

Equations (39), (45), and (51) can be represented as

$$\left(rh_\xi\right)^{-\tau} h_\eta = 1, \tag{90}$$

where $\tau = -1$ in Equation (39), $\tau = -m$ in Equation (45), and $\tau = -1/t$ in Equation (51). Equation (90) can be rewritten in the form

$$h_\xi = h \text{ and } h_\eta = (rh)^\tau. \tag{91}$$

Then, Equations (4) and (5) become

$$\frac{\partial h}{\partial \eta} + (rh)^\tau \frac{\partial \psi}{\partial \xi} = 0 \text{ and } \frac{\partial \psi}{\partial \eta} - \tau h^{\tau-2} r^\tau \frac{\partial h}{\partial \xi} = \tau r^{\tau-1} h^\tau \cos \psi. \tag{92}$$

Here, the second equation in (6) has been used to eliminate the derivative $\partial r/\partial \xi$. Using a standard technique, one can find the characteristic curves as

$$\frac{d\xi}{d\eta} = \pm\sqrt{-\tau}\, h^{\tau-1} r^\tau. \tag{93}$$

It is seen from this equation that the characteristics are real if $\tau < 0$. It is assumed that this inequality is satisfied. The characteristic coordinates are denoted as $(\alpha, \beta)$. The upper sign in (93) corresponds to $\beta$-curves and the lower sign to $\alpha$-curves. Using a standard technique, one can find the characteristic relations as

$$\begin{aligned} d\psi - \sqrt{-\tau}\frac{dh}{h} = \tau r^{\tau-1} h^\tau \cos \psi\, d\eta \text{ along the } \alpha - \text{lines}, \\ d\psi + \sqrt{-\tau}\frac{dh}{h} = \tau r^{\tau-1} h^\tau \cos \psi\, d\eta \text{ along the } \beta - \text{lines}. \end{aligned} \tag{94}$$

Equations (6), (74), and (91) yield

$$\frac{\partial r}{\partial \alpha} = h \cos \psi \frac{\partial \xi}{\partial \alpha} - (hr)^\tau \sin \psi \frac{\partial \eta}{\partial \alpha} \text{ and } \frac{\partial r}{\partial \beta} = h \cos \psi \frac{\partial \xi}{\partial \beta} - (hr)^\tau \sin \psi \frac{\partial \eta}{\partial \beta}. \tag{95}$$

The equations in (93) can be rewritten as

$$\frac{\partial \xi}{\partial \alpha} + \sqrt{-\tau}\, h^{\tau-1} r^\tau \frac{\partial \eta}{\partial \alpha} = 0 \text{ and } \frac{\partial \xi}{\partial \beta} - \sqrt{-\tau}\, h^{\tau-1} r^\tau \frac{\partial \eta}{\partial \beta} = 0 \tag{96}$$

and the equations in (94) as

$$\frac{\partial \psi}{\partial \alpha} - \frac{\sqrt{-\tau}}{h}\frac{\partial h}{\partial \alpha} = \tau r^{\tau-1} h^\tau \cos \psi \frac{\partial \eta}{\partial \alpha} \text{ and } \frac{\partial \psi}{\partial \beta} - \frac{\sqrt{-\tau}}{h}\frac{\partial h}{\partial \beta} = \tau r^{\tau-1} h^\tau \cos \psi \frac{\partial \eta}{\partial \beta}. \tag{97}$$

Equations (96) and (97), together with one of the equations in (95), constitute the system of five equations for $h$, $\psi$, $\xi$, $\eta$, and $r$. The equation for $z$ follows from (6), (74), and (91) as

$$\frac{\partial z}{\partial \alpha} = h \sin \psi \frac{\partial \xi}{\partial \alpha} + (hr)^\tau \cos \psi \frac{\partial \eta}{\partial \alpha} \text{ and } \frac{\partial z}{\partial \beta} = h \sin \psi \frac{\partial \xi}{\partial \beta} + (hr)^\tau \cos \psi \frac{\partial \eta}{\partial \beta}. \tag{98}$$

Equations (39), (46), and (52) can be treated similarly. In particular, Equation (90) should be replaced with

$$rh_\eta h_\xi^{-\tau} = 1, \tag{99}$$

where $\tau = -1$ in Equation (39), $\tau = -m$ in Equation (46), and $\tau = -1/t$ in Equation (52).

## 7. Conclusions

The present paper has reviewed and summarized the available results on geometric properties of principal stress trajectories in regions where a yield criterion is satisfied. Plane strain and axisymmetric problems have been considered. The final result for each yield criterion is a relation between the scale factors of the principal line coordinate system. Having these relations, one can reduce the boundary value problems in plasticity to purely geometric problems of finding orthogonal coordinate systems. The latter problems are represented by standard systems of partial differential equations, as described in Section 5. In most cases, these systems are hyperbolic.

The above is one of several available methods for solving boundary value problems. This method has not yet been employed for solving specific boundary value problems. Free surface problems constitute an important class of boundary value problem in plasticity [27,28]. Since a free surface coincides with an $\xi$- or $\eta$-coordinate curve, the method described in the present paper can efficiently solve free surface problems.

**Author Contributions:** Formal analysis, S.A. and M.R.; conceptualization, Y.L.; supervision, Y.L.; writing—original draft, S.A. and M.R. All authors have read and agreed to the published version of the manuscript.

**Funding:** This research has been supported by the Foreign Expert Project from the Ministry of Science and Technology of China (G2022177004L).

**Institutional Review Board Statement:** Not applicable.

**Informed Consent Statement:** Not applicable.

**Data Availability Statement:** Not applicable.

**Acknowledgments:** This publication has been supported by the RUDN University Scientific Projects Grant System, project No. 202247-2-000.

**Conflicts of Interest:** The authors declare no conflict of interest.

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
