# Peer review of "Principal Stress Trajectories in Plasticity under Plane Strain and Axial Symmetry"

_symmetry, doi:10.3390/sym15050981_

Round 1

Reviewer 1 Report

In this study, available results regarding the geometric properties of principal stress trajectories in regions where a yield criterion is satisfied are reviewed and summarized by considering plane strain and axisymmetric problems.

I would suggest that the manuscript be published as it is. However, I would expect to see the physical problem applications of the obtained formulations and to be proven to be correct.

Author Response

In this study, available results regarding the geometric properties of principal stress trajectories in regions where a yield criterion is satisfied are reviewed and summarized by considering plane strain and axisymmetric problems.

I would suggest that the manuscript be published as it is. However, I would expect to see the physical problem applications of the obtained formulations and to be proven to be correct.

RESPONSE.

The possible physical applications are outlined in the Introduction. Those are the theory of ideal flows and structural design. However, we are not in a position to review these applications, and it is not an objective of this review article. Its objective, as follows from the title of the manuscript, is to describe some mathematical properties of the principal stress trajectories. Moreover, being a review article, it does not contain any new results to be proven. All the results presented have already been proven in available publications.

Reviewer 2 Report

After studying the assessed contribution, I give the following opinion. The present paper has reviewed and summarized the available results on geometric properties of principal stress trajectories in regions where a yield criterion is satisfied. Plane strain and axisymmetric problems have been considered. The final result for each yield criterion is a relation between the scale factors of the principal line coordinate system. Having these relations, one can reduce the boundary value problems in plasticity to purely geometric problems of finding orthogonal coordinate systems.

Equations are written neatly and clearly. The symbols in the equations are well explained and have a good explanatory value. Equations (32,33,34) need to be rewritten so that they have a uniform style with the other equations. Figure (1) is appropriately classified. Comments: 1, expand and analyze in more detail the contributions used in the creation of the contribution.

2, it is necessary to emphasize and state the scientific contribution of the contribution.

Author Response

After studying the assessed contribution, I give the following opinion. The present paper has reviewed and summarized the available results on geometric properties of principal stress trajectories in regions where a yield criterion is satisfied. Plane strain and axisymmetric problems have been considered. The final result for each yield criterion is a relation between the scale factors of the principal line coordinate system. Having these relations, one can reduce the boundary value problems in plasticity to purely geometric problems of finding orthogonal coordinate systems.

Equations are written neatly and clearly. The symbols in the equations are well explained and have a good explanatory value. Equations (32,33,34) need to be rewritten so that they have a uniform style with the other equations. Figure (1) is appropriately classified. Comments: 1, expand and analyze in more detail the contributions used in the creation of the contribution.

2, it is necessary to emphasize and state the scientific contribution of the contribution.

Response.

Equations (32) – (34) have been reformatted.

Comments 1 and 2. This article is a review article, as you correctly note at the beginning of your review. Therefore, the entire article explains and summarizes the available contributions. No expansion is possible because it is not a research article. It does not provide any new results.